# How is maternal employment associated with infant and young child feeding in Bangladesh? A systematic literature review and meta-analysis

**M. A. Rifat**[1]*, **Plabon Sarkar**[2], **Israth Jahan Rimu**[3¤], **Syeda Saima Alam**[3], **Tasnu Ara**[4], **Tobias Lindström Battle**[1], **Manzur Kader**[1], **Sanjib Saha**[5]

1 Department of Global Public Health, Karolinska Institutet, Stockholm, Sweden, 2 Caritas Bangladesh, Dhaka, Bangladesh, 3 Department of Food Technology and Nutrition Science, Noakhali Science and Technology University, Noakhali, Bangladesh, 4 Nutrition International, Dhaka, Bangladesh, 5 Department of Clinical Sciences, Health Economics Unit, Lund University, Lund, Sweden

¤ Current address: Department of Nutrition and Food Engineering, Daffodil International University, Savar, Dhaka, Bangladesh

* m.a.rifat@ki.se

**Data Availability Statement:** All relevant data are within the manuscript and its Supporting Information files.

## Abstract

### Background

In the last three decades, the increasing trend in female employment in Bangladesh has been critically analyzed from a socioeconomic point of view; however, its impact on infant and young child feeding (IYCF) practices has yet to be systematically reviewed. The aim of this systematic review and meta-analysis is to investigate the association between these variables.

### Methods

A systematic literature search was conducted in PubMed, Medline, Web of Science, Embase, CINAHL, and Google Scholar to retrieve relevant records with no restriction of publication period. The Covidence tool was used for screening and data extraction. Meta-analysis was carried out using random effect models. The Newcastle-Ottawa scale was used for the quality assessment of the included articles.

### Results

A total of 24 articles were included. Of these, 16 focused on breastfeeding-related indicators, 6 focused on complementary feeding-related indicators, and 2 focused on both. Maternal employment was found to have both positive (protective) and negative (detrimental) associations with exclusive breastfeeding, whereas it was mainly positively associated with complementary feeding practices. Meta-analysis showed the pooled odds ratio of recommended early initiation of breastfeeding, exclusive breastfeeding, and complementary feeding among employed mothers were 0.79 (95% CI: 0.49, 1.27; p = 0.33), 0.32 (95% CI:0.16,

**Funding:** The author(s) received no specific funding for this work.

**Competing interests:** The authors have declared that no competing interests exist.

0.67; p = 0.002), and 1.07 (95% CI: 0.81, 1.42; p = 0.63) compared to their counterparts, respectively.

## Conclusions

Maternal employment appears not to be a protective factor for some important breastfeeding indicators in Bangladesh. For example, there was a statistically significant lower likelihood of exclusive breastfeeding practice among employed mothers as compared to those who were not employed. Therefore, these issues should be taken into consideration when formulating relevant policies and interventions, e.g., breastfeeding-friendly workplace.

## Introduction

Infant and young child feeding (IYCF) is a recommended dietary guideline for infants (0–11 months of age) and young children (12–23 months of age) which is crucial for their survival, health and well-being [1]. Broadly, in Bangladesh, IYCF consists of components which include breastfeeding within one hour of birth or early initiation of breastfeeding (EIBF), exclusive breastfeeding (EBF) until six months of age, initiation of recommended complementary feeding after six months of age, and continuation of breastfeeding at least until two years of age [2]. Sub-optimum IYCF is a major determinant of childhood malnutrition and a serious public health concern particularly in low- and lower middle-income countries including Bangladesh [3]. In Bangladesh, 65% of infants aged less than 6 months are exclusively breastfed and only 34% of children aged 6–23 months receive recommended complementary feeding, demonstrating higher exclusive breastfeeding rate than India (43%) and Nepal (57%) [4, 5]. Meanwhile, 31% of children aged under five years are stunted, and 22% are underweight. These proportions are similar with neighboring countries, including India (36% stunting and 32% underweight) and Nepal (25% stunting and 19% underweight) [6, 7]. The under-5 mortality rate is 45 deaths per 1000 live births, and the infant mortality rate is 38 deaths per 1000 live births according to Bangladesh Demographic and Health Survey (BDHS) 2017–18 [4]. Improving IYCF practices may be an important factor in reducing undernutrition and child mortality [8–10].

Parental involvement in IYCF practices differs by context. In Bangladesh, child feeding is usually performed by mothers [11], whereas family food choice is primarily decided by fathers [12] Therefore, child feeding at the family level, especially complementary feeding, is mostly dependent on paternal food preference [12]. However, this may be different where mothers are in employment. Employed mothers can independently spend earnings on food for their children and therefore diversify food intake, regardless of their husbands' income [13]. Studies conducted in lower middle-income countries (LMICs) have shown that employed mothers have better knowledge of nutrition and child feeding, can more easily access and interpret nutritional information [14], and make use of available health services [15]. These abilities could entail better IYCF practices [14, 16, 17]. However, while maternal employment could lead to better complementary feeding, it may simultaneously have a negative effect on breastfeeding practices. A study using nationally representative data from Bangladesh has found that maternal employment could lead to increased use of breastmilk substitutes, bottle feeding, and early cessation of breastfeeding [18]. Furthermore, employed mothers might breastfeed insufficiently due to long working hours [19, 20]. Sub-optimum policy support related to maternity leave or lack of breastfeeding-friendly workplaces, including breastfeeding breaks, can lead to

poor breastfeeding practices among employed mothers [21]. For example, in Bangladesh, six months fully paid maternity leave, which is a legally protected right, is not strictly implemented in all job sectors, and not all workplaces are adequately equipped to provide breastfeeding-friendly environment [22].

There has been a significant improvement in access to adequate nutrition in Bangladesh over the last three decades [19]. Research suggests that this is largely due to changes in socio-economic status of the population, in part thanks to income and household wealth accumulation [23]. Women's participation in the workforce was a catalytic factor of this change [24, 25]. However, the association between maternal employment status and IYCF practices has yet to be reviewed. This systematic review aims to fill this knowledge gap by understanding the influences of maternal employment on IYCF practices. Meta-analysis will provide a consensus of the individual associations between maternal employment and IYCF practices over the years. This could facilitate the implementation of IYCF policies and programs for both employed and unemployed mothers in Bangladesh to improve IYCF practices.

## Methods

This systematic review was registered to PROSPERO with registration ID CRD42023408507. Preferred Reporting Items for Systematic Review and Meta-Analysis (PRISMA) was used as the reporting guideline (S1 Appendix) [26].

### Data sources and search strategy

Systematic literature search strategies were developed for PubMed, Web of Science, Medline, CINAHL, and Embase. Keywords were used to conduct literature search in Google Scholar. The literature search in the databases was carried out from 10 January 2023 to 26 March 2023. The search strategy has been provided in the S1 Appendix.

### Inclusion and exclusion criteria

The PECO framework was used to determine record inclusion as follows: population (P) = mothers having an infant (0–11 months of age) or a young child (12–23 months of age) in Bangladesh; exposure (E) = mothers being employed in any public or private sector jobs or self-employed; comparator (C) = mothers without employment; and outcome (O) = status of IYCF practices, e.g., EIBF, EBF, breastfeeding duration, and quality of complementary feeding. Only observational studies were considered for inclusion because the current study aims to observe the association between the maternal employment and IYCF practices in real-world settings. There was no limit on the publication period, and only records published in English were considered. Records that did not fulfill the inclusion criteria were excluded.

### Selection process

After conducting literature searches in the selected databases, title and abstract of the records were imported to Covidence (https://www.covidence.org/) for duplicate removal and screening. The title and abstract were screened by at least two of the six reviewers (FA, TA, IJR, SSA, PS, and MAR), and conflicts were independently resolved by a third reviewer (MAR, PS, and SS). The full text of each record qualifying the title and abstract screening was reviewed by two reviewers, and a third reviewer independently resolved any conflicts. Articles fulfilling full text screening criteria were included for review.

## Data extraction

For each of the included records, data was extracted by two of the five reviewers working independently (TA, IJR, SSA, PS, and MAR). One reviewer (MAR) conducted cross-checking of the extracted data and resolved conflicts through mutual discussion. The following variables were included in the data extraction: identity of the records, study population, geographical area covered, study design, data source, IYCF indicators considered, maternal employment status, type of data analysis carried out, effect size, confidence interval, p-value, and variables adjusted for during analysis.

## Data synthesis and meta-analysis

Meta-analysis was carried out in relation to three IYCF domains. These were 1) early initiation of breastfeeding, 2) exclusive breastfeeding until six months of age as recommended by WHO, and 3) complementary feeding. Complementary feeding indicators included were minimum dietary diversity (MDD), referring to the presence of food items from at least five out of eight different food groups, and minimum acceptable diet (MAD), consisting of MDD plus recommended daily frequency of the diet [27]. Other indicators including bottle feeding, ever breastfeeding, or provision of iron rich complementary foods were not considered in meta-analysis due to scarcity of sufficient evidence. Effect estimates and their 95% confidence intervals reported in the studies were considered for meta-analysis. The odds ratio (OR) indicating the likelihood of recommended IYCF practices among employed mothers as compared to unemployed mothers was synthesized where records only presented the frequency distribution of the considered variables. Data was also synthesized where the effect estimate was given with an indicator other than odds ratio, rate ratio (RR), or hazard ratio (HR). Where the same data source was used in more than one article to demonstrate the association between the same variables of interest, only one of these articles was selected for the meta-analysis to avoid duplication bias (this was frequently the case where Bangladesh Demographic and Health Survey (BDHS) had been used). If several studies were conducted based on the same data set, the study considering more relevant variables adjusted in the statistical model was included. Furthermore, where the same dataset was used for both MDD and MAD, inclusion of MAD was favored as it is a richer measure of complementary feeding practices. Random effect modelling was used for meta-analysis to account for differences in the association between the variables of interest based on the time-period of the study and the study population. In addition, subgroup analyses based on the national representativeness of the study population were carried out to observe variations in heterogeneity. Furthermore, leave-one-out meta-analysis was carried out to observe how the pooled effect size was affected due to the omission of any of the included studies [28]. A meta funnel plot was produced to observe the small study effect or publication bias [29]. The statistical software package STATA (version 17 StataCorp, College Station, Texas) was used for conducting the meta-analysis.

## Assessment of quality and risk of bias

The Newcastle-Ottawa Scale (NOS) was used to assess the quality of the included articles [30]. The NOS scale has different criteria to assess both descriptive (cross-sectional) and analytical (case-control and cohort) studies. To address the risk of bias in the selection process, each record was screened independently by at least two reviewers and conflicts were resolved by a third reviewer (MAR) through collective discussion. The quality of the included articles was further assessed by at least two additional independent reviewers, and conflicts were resolved by another reviewer (MAR) through collective discussion.

## Terminologies

Maternal employment was defined as maternal involvement in any income-generating activity as mentioned by the authors of the original articles. An association was considered to be positive when maternal employment was found to be protective of IYCF practices and negative when maternal employment was detrimental to IYCF practices.

## Ethical considerations

This study included published articles as sources of data; therefore, informed consent and getting ethical approval were not required. However, the presence of publication bias remains as an ethical concern for this study because it could mislead the pooled effect estimates. Despite potential risk, this study systematically summarized all relevant evidence on how the maternal occupation was associated with IYCF. The findings may be of importance at the policy level. Furthermore, to conduct this study, validated tools such as PRISMA as reporting guideline were used. Therefore, considering the strength of the methods and the public health importance of the topic, the potential benefits might outweigh the potential risk.

## Results

A total of 1,917 records were retrieved for screening. Of these, 24 studies met the inclusion criteria and were reviewed. The selection process for the included studies is presented in Fig 1.

### Characteristics of the included studies

The characteristics of the included studies have been summarized in Table 1.

The publication period of the included studies was from 1981 to 2022. A total of 23 studies were cross-sectional and one was longitudinal. Fifteen were conducted at the national level, five were conducted at the district level [38–40, 44, 48], and four were conducted at the sub-district level [16, 35, 46, 49]. Of the five district-level studies, two were conducted at hospitals in Dhaka [39, 40]. Fourteen of the studies utilized data from the Bangladesh Demographic and Health Survey.

In the majority of the included studies, mothers were categorized as employed versus unemployed. Only two studies specified the type of maternal employment [16, 44]. Of these, one study categorized maternal employment as skilled and unskilled [16], while the second categorized employment as governmental employment, private employment, and no employment [44].

The association between maternal employment and IYCF indicators is presented in Tables 2 and 3.

### Maternal employment and breastfeeding indicators

**Early Initiation of Breastfeeding (EIBF).**   Four studies focused on the early initiation of breastfeeding [31, 40, 45, 47]. Of these, three found that employed mothers were less likely to initiate early breastfeeding as compared to unemployed mothers [31, 40, 45]. One study found the opposite association [47].

**Exclusive Breastfeeding (EBF).**   Eleven studies [20, 21, 31, 32, 35, 37–39, 45, 48, 49], published from the analysis of 11 unique datasets/surveys, reported the associations between maternal employment and exclusive breastfeeding. Of these, ten found that employed mothers were less likely to exclusively breastfeed their children as compared to unemployed mothers [20, 21, 31, 32, 35, 37–39, 45, 49] whereas one study found the opposite [48].

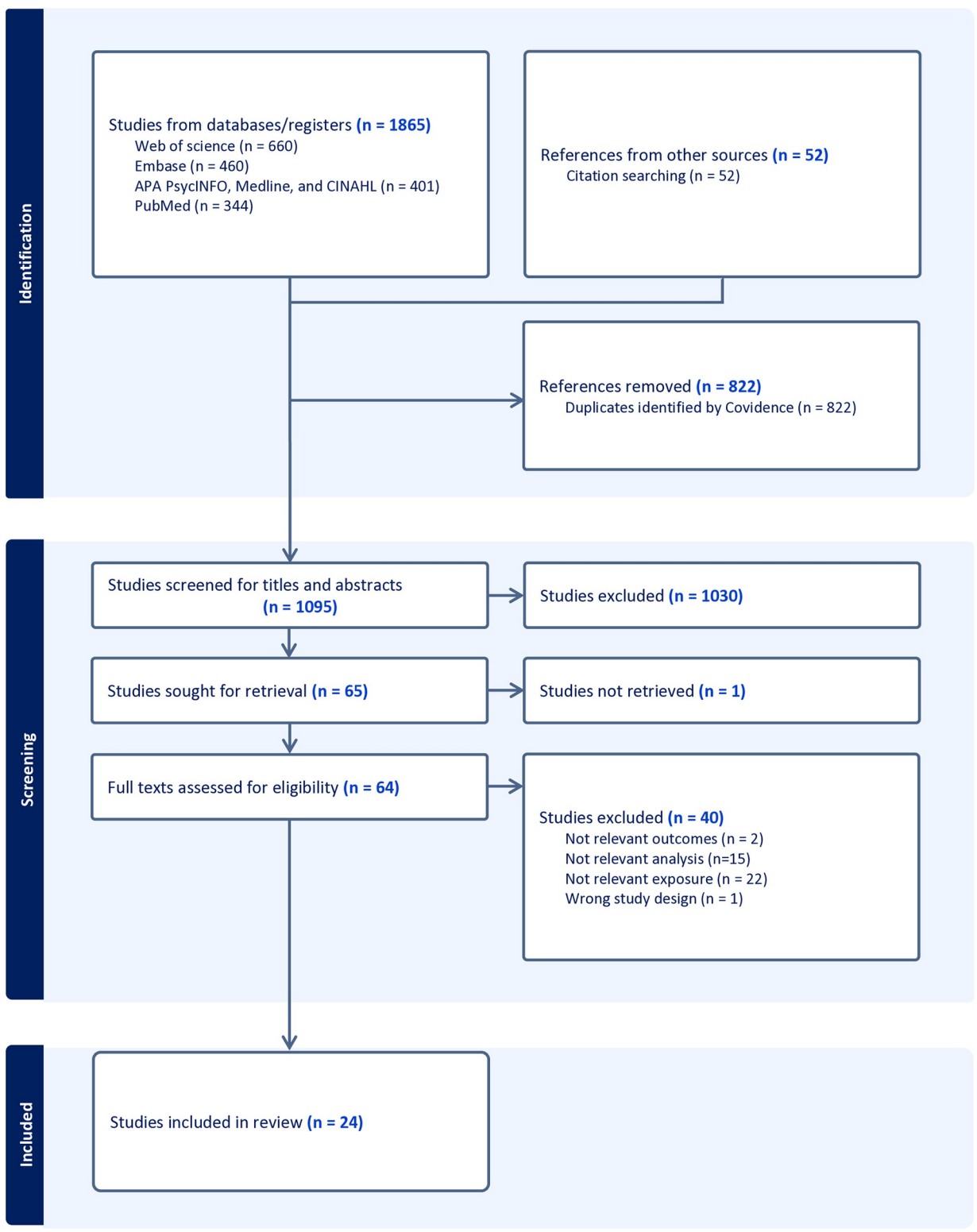

**Fig 1. Selection of the included studies.**

**Table 1. Characteristics and quality of the included studies.**

| Reference | Author, year of publication | Study design | Geographical area covered | Data source | Maternal age (year) | Quality score[a] |
|---|---|---|---|---|---|---|
| [31] | Ahmed, 2022 | Cross-sectional | National | BDHS[b] 2004–2018 | 15–49 | 10 |
| [32] | Ahmmed, 2022 | Cross-sectional | National | BDHS 2011–2018 | 15–49 | 10 |
| [18] | Akter, 2010 | Cross-sectional | National | BDHS 2004 | ≤ 24: 48.98%<br>25–34: 40.53%<br>≥ 35: 10.49% | 08 |
| [33] | Akter, 2010 | Cross-sectional | National | BDHS 2004 | 10–49 | 08 |
| [16] | Ali, 2019 | Cross-sectional | 20 sub-districts | Household survey 2014–2017 | 15–49 | 10 |
| [34] | Ayesha, 2021 | Cross-sectional | National | BDHS 2014 | 15–49 | 10 |
| [35] | Basnet, 2020 | Cross-sectional | 20 sub-districts | Alive and Thrive baseline data 2010 | Not reported | 10 |
| [36] | Blackstone, 2018 | Cross-sectional | National | BDHS 2011 and 2014 | 12–49 | 10 |
| [37] | Blackstone, 2018 | Cross-sectional | National | BDHS 2007, 2011 and 2014 | 12–49 | 10 |
| [38] | Haider, 2019 | Longitudinal | Chattogram | Survey 2015–2017 | 18–30 | 07 |
| [39] | Hasan, 2021 | Cross-sectional | Dhaka district | Survey 2019 | Mean: 26.1 ± 5.3 | 09 |
| [40] | Hasan, 2020 | Cross-sectional | Dhaka district | Survey 2019 | Mean: 27.5 ± 5.1 | 08 |
| [20] | Hossain, 2018 | Cross-sectional | National | BDHS 2014 | 15–49 | 10 |
| [41] | Kabir, 2012 | Cross-sectional | National | BDHS 2007 | 15–49 | 10 |
| [42] | Jain, 1981 | Cross-sectional | National | World Fertility Surveys 1976 | 15–40: 98.6%<br>> 40: 1.4% | 10 |
| [43] | Khan, 2019 | Cross-sectional | National | BDHS 2011, 2014 | 12–49 | 08 |
| [44] | Kundu, 2022 | Cross-sectional | 6 districts | Survey 2021 | ≤ 20: 19.24%<br>20–30: 69.33%<br>>30: 11.43% | 08 |
| [45] | Mihrshahi, 2010 | Cross-sectional | National | BDHS 2004 | 10–49 | 10 |
| [46] | Nguyen, 2013 | Cross-sectional | 20 sub-districts | Alive and Thrive baseline data 2010 | Mean: 26.4 ± 6 | 10 |
| [21] | Rahman, 2020 | Cross-sectional | National | BDHS 2011 and 2014 | 15–49 | 10 |
| [47] | Raihana, 2021 | Cross-sectional | National | BDHS 2014 | 15–49 | 08 |
| [48] | Rana, 2020 | Cross-sectional | Rural Rajshahi | Survey 2015 | ≤ 20: 61.2%<br>≥ 21: 38.8% | 10 |
| [49] | Rasheed, 2009 | Cross-sectional | Matlab | Survey[c]<br>2002–2004 | Mean: 26.2 ± 4.3 | 08 |
| [2] | Sheikh, 2019 | Cross-sectional | National | BDHS 2014 | Mean: 23.99 ± 5.54 | 10 |

[a]Quality score was assessed by Newcastle-Ottawa Scale, 9–10: very good, 7–8: good, 5–6: satisfactory, <5: unsatisfactory (S1 Appendix).

[b]Bangladesh Demographic and Health Survey

[c]Maternal and Infant Nutrition Intervention in Matlab (MINIMat). In all studies, mothers with different levels of educational attainments, ranging from no education to tertiary level education, were included.

**Continued breastfeeding.** Of the three studies investigating the association of maternal employment and breastfeeding duration, a positive association was reported in two studies [31, 34] and the other study reported a negative association [42].

**Cessation of breastfeeding and bottle-feeding.** Unemployed mothers represented an earlier breastfeeding cessation than that of their counterparts according to all three studies included in the review [18, 33, 43]. Meanwhile, one study [45] investigated the association between maternal employment status and bottle-feeding practices and found that employed mothers had significantly higher odds (OR: 1.88, 95% CI: 1.19, 2.98) of bottle feeding (usually infant formula, powdered milk, and cow's milk) their children compared to the unemployed mothers.

**Table 2. Association between maternal employment and breastfeeding-related indicators (early initiation of breastfeeding, exclusive breastfeeding, breastfeeding continuation, bottle feeding, colostrum feeding, pre-lacteal feeding, and feeding breastmilk substitutes) according to included studies.**

| Study | Breastfeeding indicators | Sample size[a] | Outcome definition | Employment status or categories | Association | Confidence interval | P-value | Type of analysis | Variables adjusted for |
|---|---|---|---|---|---|---|---|---|---|
| Ahmed, 2022 [31] | Early Initiation of breastfeeding (BDHS 2017–18) | 2137 | Providing early initiation of breastfeeding | No[b] | 1.00 | - | - | Multilevel multivariable logistic regression | Childbirth order, mother's age, mother's education, number of antenatal care visits, place of delivery, mode of delivery, wealth index, sex of the household head, type of place of residence, division |
| | | | | Yes | 0.93 | 0.78, 1.10 | ≥ 0.05 | | |
| | Exclusive breastfeeding (BDHS 2017–18) | 665 | Providing exclusive breastfeeding | No[b] | 1.00 | - | - | Multilevel multivariable logistic regression | Mother's age, place of delivery, family size, type of place of residence, division |
| | | | | Yes | 0.68 | 0.48, 0.95 | < 0.05 | | |
| | Continued breastfeeding (BDHS 2017–18) | 489 | Continuation of breastfeeding at 2 years of age | No[b] | 1.00 | - | - | Multilevel multivariable logistic regression | Childbirth order, number of antenatal care visits, mode of delivery, wealth index, sex of the household head, type of place of residence, division |
| | | | | Yes | 1.24 | 0.71, 2.16 | ≥ 0.05 | | |
| Ahmmed, 2022 [32] | Exclusive breastfeeding | 2317 | The odds of providing exclusive breastfeeding | Unemployed[b] | 1.00 | - | - | Three-level logistic regression | Year, mothers' current age, sex of child, mother's education, wealth index, initiation of breastfeeding |
| | | | | Employed | 0.76 | 0.59, 0.96 | 0.03 | | |
| Akter, 2010 [18] | Cessation of breastfeeding | 5366 | The relative risk of cessation of breastfeeding | Work[b] | 1.00 | - | - | Cox's proportional hazard model | Residence, region, religion, educated level, work status, mother's age, age at marriage, parity, use of contraceptives |
| | | | | Doesn't work | 1.14 | 1.06, 1.23 | 0.00 | | |
| Akter, 2010 [33] | Cessation of breastfeeding | 5455 | The relative risk of cessation of breastfeeding | Work[b] | 1.00 | - | - | Cox's proportional hazard model | Age of mothers, age-at-marriage, sex of the child, parity, contraceptives-se, delivery status, place of residence, division/region, religion, educational level, work status, education of husband |
| | | | | Do not work | 1.16 | 1.08, 1.25 | < 0.01 | | |
| Ayesha, 2021 [34] | Breastfeeding duration | 3541 | Duration of breastfeeding | Housewife[b] | OR = 1.00 | - | - | Multiple linear regression | Antenatal care visit, mode of delivery, place of delivery, mother's age at first birth, total children ever born, mother's age, mother's educational level, father's educational level, division, religion, wealth index, mother's body mass index |
| | | | | Working outside of house | 1.24[c] (OR = 3.46) | 0.63, 1.85 (1.88, 6.36) | 0.001 | | |
| Basnet, 2020 [35] | Exclusive breastfeeding | 977 | The odds of providing exclusive breastfeeding | Unemployed[b] | 1.00 | - | - | Multiple logistic regression | Education, knowledge, height, well-nourished, mental well-being, decision-making, support in domestic tasks, perceived instrumental support |
| | | | | Employed | 0.69 | Not reported | ≥ 0.05 | | |

(*Continued*)

**Table 2.** (Continued)

| Study | Breastfeeding indicators | Sample size[a] | Outcome definition | Employment status or categories | Association | Confidence interval | P-value | Type of analysis | Variables adjusted for |
|---|---|---|---|---|---|---|---|---|---|
| Blackstone, 2018 [37] | Exclusive breastfeeding (2007) | 515 | Providing exclusive breastfeeding | Not employed[b] | OR = 1.00 | - | - | Multilevel binary logistic regression | Gender of child, age of child |
| | | | | Employed | -0.29[c] (OR = 0.75) | -0.96, 0.38 (0.38, 1.46) | ≥ 0.05 | | |
| | Exclusive breastfeeding (2011) | 800 | Providing exclusive breastfeeding | Not employed[b] | OR = 1.00 | - | - | Multilevel binary logistic regression | Gender of child, age of child |
| | | | | Employed | 0.47[c] (OR = 1.60) | -0.37, 1.3 (0.69, 3.67) | ≥ 0.05 | | |
| | Exclusive breastfeeding (2014) | 638 | Providing exclusive breastfeeding | Not employed[b] | OR = 1.00 | - | - | Multilevel binary logistic regression | Gender of child, age of child |
| | | | | Employed | 0.15[c] (OR = 1.16) | -0.44, 0.73 (0.64, 2.08) | ≥ 0.05 | | |
| Haider, 2019 [38] | Exclusive breastfeeding | 304 | The odds of providing exclusive breastfeeding | Not employed[b] | 1.00 | - | - | Odds ratio synthesized from raw data | Not applicable |
| | | | | Employed | 0.34 | 0.11, 0.87 | 0.02 | | |
| Hasan, 2021 [39] | Exclusive breastfeeding | 385 | The odds of providing exclusive breastfeeding | Service holder[b] | 1.00 | - | - | Multiple logistic regression | Maternal age, maternal education level, family monthly income (BDT), family type, marital status of mother, live birth, place of delivery, mode of delivery |
| | | | | Housewife | 5.84 | 2.42, 14.13 | < 0.001 | | |
| Hasan, 2020 [40] | Early initiation of breastfeeding | 422 | The odds of early initiation of breastfeeding for neonates | Service holder[b] | 1.00 | - | - | Multiple logistic regression | Maternal age, maternal education, family income (BDT), live birth, types of delivery, ANC check-up, pre-lacteal foods given, counseling before delivery, baby's birth weight, skin to skin contact after delivery |
| | | | | Housewife | 2.42 | 1.39, 4.23 | 0.002 | | |
| Hossain, 2018 [20] | Exclusive breastfeeding | 3541 | The odds of providing exclusive breastfeeding | Service holder[b] | 1.00 | - | - | Multivariate binary logistic regression | Region, mother's age, mother's education, father's education, father's occupation, mass media access, mother's BMI, total children ever born, delivery mode for last pregnancy, delivery place, antenatal care, postnatal care for mother, breastfeeding counseling during first two days, current age of children |
| | | | | Housewife | 1.20 | 1.02, 1.42 | 0.05 | | |
| Jain, 1981 [42] | Duration of breastfeeding | 2660 | Duration of breastfeeding | Not working[b] | OR = 1.00 | Not reported | Not reported | Multiple linear regression | Age, parity, infant death, mother's education, residence, sex of child, mother's workplace |
| | | | | Working | -0.12[c] (OR = 0.89) | Not reported | Not reported | | |
| Khan, 2019 [43] | Cessation of breastfeeding | 9278 | The risk of cessation of breastfeeding | No[b] | 1.00 | - | - | Cox's proportional hazards test | Sex, maternal age at childbirth, maternal education, paternal education, maternal working status, contraceptive use, maternal malnutrition, birth order, wealth index, religion, exposure to media, heard about family planning in media, place of residence, division/ region, survey year |
| | | | | Yes | 0.92 | Not reported | ≥ 0.05 | | |

(*Continued*)

**Table 2.** (Continued)

| Study | Breastfeeding indicators | Sample size[a] | Outcome definition | Employment status or categories | Association | Confidence interval | P-value | Type of analysis | Variables adjusted for |
|---|---|---|---|---|---|---|---|---|---|
| Rahman, 2020 [21] | Exclusive breastfeeding | 1440 | The odds of providing exclusive breastfeeding | Not working[b] | 1.00 | - | - | Multilevel logistics regression | Survey's year, mother's age, mother's educational status, partner's educational level, wealth index, place of delivery, antenatal visit, postnatal care visit, child's gender, child's age, children ever born, partner's occupation |
| | | | | Working | 0.70 | 0.48, 0.99 | < 0.05 | | |
| Raihana, 2021 [47] | Breastfeeding initiation beyond the first hour of birth | 785 | The odds of early initiation of breastfeeding beyond one hour after birth | Employed[b] | 1.00 | - | - | Univariate logistic regression | Unadjusted |
| | | | | Not employed | 1.10 | 0.70, 1.74 | 0.67 | | |
| Rana, 2020 [48] | Exclusive breastfeeding | 513 | The odds of providing exclusive breastfeeding | Housewife[b] | 1.00 | - | - | Binary logistic regression | Age in years, religion, delivery place, educational status, type of family, monthly family income in BDT |
| | | | | Service holder | 9.99 | 4.49, 22.26 | < 0.01 | | |
| Rasheed, 2009 [49] | Exclusive breastfeeding | 810 | The odds of providing exclusive breastfeeding | Mother not employed[b] | 1.00 | - | - | Logistic regression | Pregnancy food, micronutrient supplementation |
| | | | | Mother employed | 0.49 | 0.19, 1.32 | ≥ 0.05 | | |
| Mihrshahi, 2010 [45] | Bottle feeding | 1263 | The odds of providing bottle feeding | No[b] | 1.00 | - | - | Multiple logistic regression | Mother's marital status, maternal education, husband's education, mother's age, birth order of child, preceding birth interval, sex of baby, age of child, place of delivery, mode of delivery, type of delivery assistance, number of antenatal clinic visits, timing of postnatal checkup, mother's BMI, mother reads newspapers, mother listens to radio, mother watches television, household wealth index, number of categories of decisions in which women have final say, residence, geographic region |
| | | | | Yes | 1.88 | 1.19, 2.98 | < 0.01 | | |
| | Not exclusive breastfeeding | 677 | The odds of not exclusively breastfeeding | No[b] | 1.00 | - | - | Univariate analysis | Unadjusted |
| | | | | Yes | 1.94 | 1.10, 3.44 | 0.023 | | |
| | Not early initiation of breastfeeding | 1263 | The odds of non-early initiation of breastfeeding | No[b] | 1.00 | - | - | Univariate analysis | Unadjusted |
| | | | | Yes | 1.03 | 0.66, 1.62 | Not reported | | |

Note: Depending on the type of analysis, adjusted odds ratio (aOR) and adjusted hazard ratio (aHR) were used to compare mothers with different employment categories, unless otherwise indicated

[a]Sample size corresponds to total number of mothers in respective comparison groups

[b]Reference category for comparison

[c]Regression coefficient; Confidence intervals are at 95% level unless otherwise indicated.

**Table 3. Association between maternal employment and complementary feeding-related indicators (weaning, minimum dietary diversity, minimum acceptable diet, minimum meal frequency, and introduction of solid, semi-solid and iron rich foods) according to included studies.**

| Study | Complementary feeding indicators | Sample size[a] | Outcome definition | Employment status or categories | Association | Confidence interval | P-value | Type of analysis | Variables adjusted for |
|---|---|---|---|---|---|---|---|---|---|
| Ali, 2019 [16] | Minimum dietary diversity | 6468 | Providing minimum dietary diversity | Homemaker[b] | OR = 1.00 | - | - | Multivariate linear regression | Sex, mother's age, mother's education, households' monthly expenditure on food (in Taka), households food security status, wealth quintile. |
| | | | | Unskilled worker | 0.03[c] (OR = 1.03) | Not reported | 0.74 | | |
| | | | | Skilled worker | 0.48[c] (OR = 1.62) | Not reported | <0.00 | | |
| Basnet, 2020 [35] | Minimum meal frequency | 1211 | The odds of providing minimum meal frequency | Unemployed[b] | 1.00 | - | - | Multiple regression analysis | Education, knowledge, height, well-nourished, mental well-being, decision-making power, support in domestic work, perceived instrumental support. |
| | | | | Employed | 1.80 | Not reported | < 0.05 | | |
| | Minimum dietary diversity | 1211 | The odds of providing minimum dietary diversity | Unemployed[b] | OR = 1.00 | - | - | Multiple regression analysis | Education, knowledge, height, well-nourished, mental well-being, decision-making power, support in domestic work, perceived instrumental support. |
| | | | | Employed | 0.01[c] (OR = 1.01) | Not reported | ≥ 0.05 | | |
| Blackstone, 2018 [36] | Minimum dietary diversity (2011) | 2264 | The odds of providing minimum dietary diversity | Unemployed[b] | 1.00 | - | - | Binary logistic regression | Education, wealth index, frequency of television-watching, frequency of newspaper-reading, decision-making power, place of delivery, number of antenatal visits, region of residence, infant age group. |
| | | | | Employed | 0.73 | 0.58, 1.27 | ≥ 0.05 | | |
| | Minimum dietary diversity (2014) | 2238 | The odds of providing minimum dietary diversity | Unemployed[b] | 1.00 | - | - | Binary logistic regression | Education, wealth index, frequency of television-watching, frequency of newspaper-reading, decision-making power, place of delivery, number of antenatal visits, region of residence, infant age group. |
| | | | | Employed | 1.21 | 0.95, 1.54 | ≥ 0.05 | | |
| | Complementary feeding (2011) | 661 | The odds of providing complementary feeding | Unemployed[b] | 1.00 | - | - | Binary logistic regression | Education, wealth index, frequency of television-watching, frequency of newspaper-reading, decision-making power, place of delivery, number of antenatal visits, region of residence. |
| | | | | Employed | 1.06 | 0.58, 1.77 | ≥ 0.05 | | |
| | Complementary feeding (2014) | 670 | The odds of providing complementary feeding | Unemployed[b] | 1.00 | - | - | Binary logistic regression | Education, wealth index, frequency of television-watching, frequency of newspaper-reading, decision-making power, place of delivery, number of antenatal visits, region of residence. |
| | | | | Employed | 1.20 | 0.98, 1.47 | ≥ 0.05 | | |

*(Continued)*

**Table 3.** (Continued)

| Study | Complementary feeding indicators | Sample size[a] | Outcome definition | Employment status or categories | Association | Confidence interval | P-value | Type of analysis | Variables adjusted for |
|---|---|---|---|---|---|---|---|---|---|
| Kabir, 2012 [41] | Minimum acceptable diet | 1727 | The odds of minimum acceptable diet to | No occupation | 1.00 | - | - | Odds ratio synthesized from raw data | Not applicable |
| | | | | Occupation | 1.26 | 0.99, 1.60 | 0.05 | | |
| Kundu, 2022 [44] | Minimum dietary diversity | 1190 | The odds of providing minimum dietary diversity | Government job[b] | 1.00 | - | - | Two-level logistic regression | Residence, religion, family monthly income (BDT), mother's age, mother's education, father's age, father's education, father's occupation, family size, children <5y old, order of children, children's age, sex of children, household food security, received microcredit loan, current microcredit loan. |
| | | | | Private job | 0.40 | 0.04, 3.82 | 0.42 | | |
| | | | | Housewife | 0.37 | 0.04, 3.37 | 0.38 | | |
| Mihrshahi, 2010 [45] | Not given timely complementary feeding | 401 | The odds of not providing timely complementary feeding | No[b] | 1.00 | - | - | Univariate analysis | Unadjusted |
| | | | | Yes | 1.58 | 0.76, 3.32 | ≥ 0.05 | | |
| Nguyen, 2013 [46] | Minimum dietary diversity | 1211 | The odds of providing minimum dietary diversity | Do not earn[b] | 1.00 | - | - | Multivariable logistic regression | Maternal MDD, mother's age, maternal education, maternal knowledge, maternal self-perceived physical health, mother can decide to buy foods, food security, SES, mother as household head, number of children <5 y, child age, child sex. |
| | | | | Earn | 0.64 | 0.31, 1.30 | ≥ 0.05 | | |
| | Minimum dietary diversity | 1211 | Providing minimum dietary diversity | Do not earn[b] | OR = 1.00 | - | - | Multivariable linear regression | Maternal dietary diversity, mother's age, maternal education, maternal knowledge, maternal self-perceived physical health, mother can decide to buy foods, food security, SES, mother as household head, number of children <5 y, children age, child sex. |
| | | | | Earn | -0.10[c] (OR = 0.91) | -0.36, 0.16 (0.68, 1.17) | ≥ 0.05 | | |
| Sheikh, 2019 [2] | Minimum acceptable diet | 2331 | The odds of providing minimum acceptable diet | Not working[b] | 1.00 | - | - | Multivariable logistic regression | Child age, sex of children, mother's age, mother's education level, birth order, access to electronic media, household size, location of residence, wealth index, division. |
| | | | | Working | 1.22 | 0.95, 1.56 | ≥ 0.05 | | |

Note: Depending on type of analysis, adjusted odds ratio (aOR) and adjusted hazard ratio (aHR) were used to compare mothers with different employment categories, unless otherwise indicated

[a]Sample size corresponds to total number of mothers in respective comparison group

[b]Reference category for comparison

[c]Regression coefficient; Confidence intervals are at 95% level unless otherwise indicated.

## Maternal occupation and complementary feeding (CF)

**Complementary feeding.** Two studies [36, 45] investigated the influence of maternal working status on overall complementary feeding practices and found that employed mothers were more likely to provide recommended complementary feeding than their unemployed counterparts.

**Minimum dietary diversity.** Of five studies [16, 35, 36, 44, 46] investigating the association between maternal employment and MDD, two reported positive association [16, 35] and two reported negative associations [36, 46]. Interestingly, one study compared how ensuring MDD varied among mothers with different employment status and found that proportion of mothers who provided their children with complementary food with MDD was higher among those with government jobs than that of those with private sector jobs [44].

**Minimum acceptable diet.** Two studies assessing the association between maternal employment and MAD of their children's complementary food found a positive association in both cases [2, 41].

## Meta-analysis

In total, four studies observing the early initiation of breastfeeding among employed mothers compared to their counterparts were included in the meta-analysis. Similarly, seven studies involving exclusive breastfeeding, and five studies involving complementary feeding were included. Analysis found the pooled odds ratios for providing early initiation of breastfeeding, exclusive breastfeeding, and complementary feeding among employed mothers to be 0.79 (95% CI: 0.49, 1.27; p = 0.33), 0.32 (95% CI: 0.16, 0.67; p = 0.002), and 1.07 (95% CI: 0.81, 1.42; p = 0.63), respectively, highlighting a statistically significant lower likelihood of exclusive breastfeeding among employed mothers compared with those who were not employed (Figs 2–4). Subgroup analysis demonstrated a significant difference in pooled association for exclusive breastfeeding between national and sub-national level studies (p = 0.02); however, such difference was not statistically significant (p = 0.95) for complementary feeding indicator (Figs 3 and 4). A high level of heterogeneity was found across all three meta-analyses ($I^2 = > 80\%$). Furthermore, publication bias was detected among the studies included in the meta-analysis (S1 Appendix). As identified by leave-one-out meta-analysis, the stability of the overall effect of maternal occupation on EIBF and CF were found to be affected by individual effect, indicating a weak stability of the pooled effect size; however, such effect was not observed for EBF (S1 Appendix).

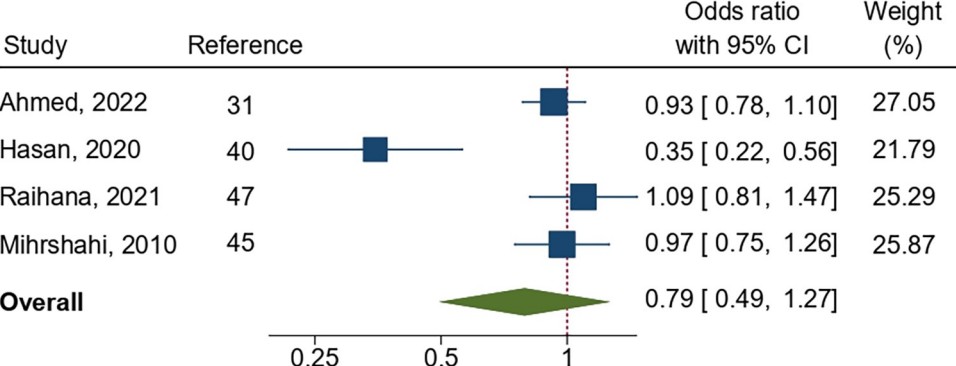

**Fig 2. Pooled odds ratio (OR) showing the likelihood of early initiation of breastfeeding among employed mothers in Bangladesh as compared to their counterparts.**

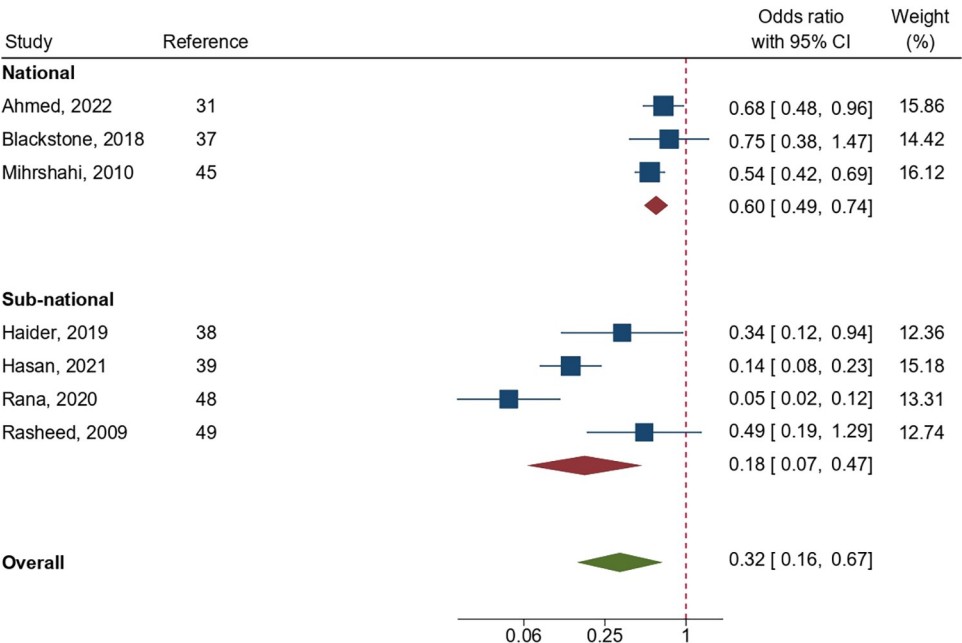

**Fig 3. Pooled odds ratio (OR) showing the likelihood of exclusive breastfeeding among employed mothers in Bangladesh as compared to their counterparts.**

## Quality of included articles

Quality assessment scores, as determined according to Newcastle-Ottawa Scale, of the included cross-sectional studies ranged from 8–10 on a scale from 0 to 10 (Table 1, S1 Appendix). Of the cross-sectional studies, 16 studies were categorized as 'very good', and 7 studies were categorized as 'good' quality studies. The quality assessment score of the included longitudinal

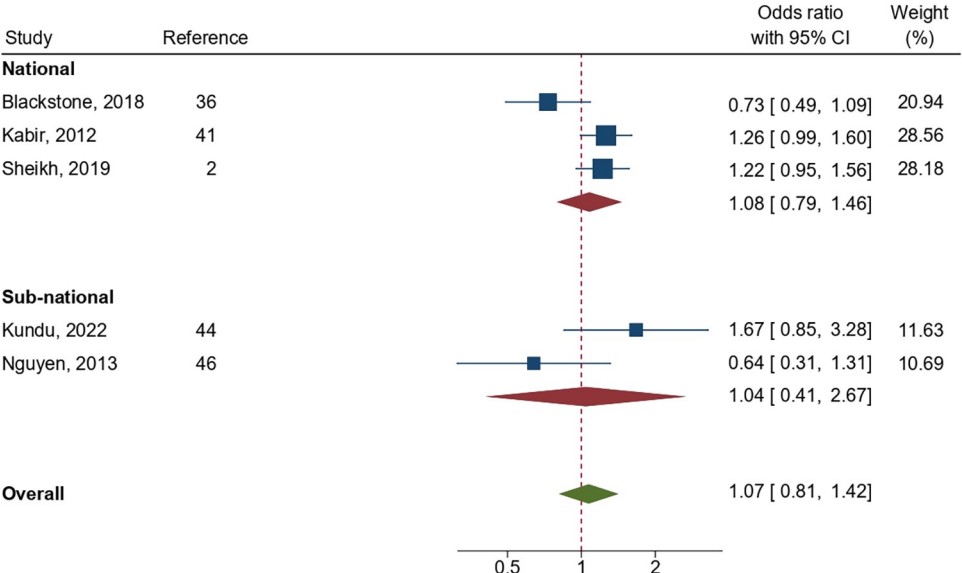

**Fig 4. Pooled odds ratio (OR) showing the likelihood of recommended complementary feeding among employed mothers in Bangladesh as compared to their counterparts.**

study was 7, which is considered 'good' (S1 Appendix). In addition to the Newcastle-Ottawa Scale, we checked each of the included studies for reporting of multicollinearity. Four studies [33, 41, 43, 44] reported standard errors and four other studies [16, 32, 47, 49] reported variance inflation factor (VIF). In addition, five studies [20, 34, 42, 46, 48] calculated the predictability ($R^2$) of the statistical model used. Of these five studies, all but one showed high predictability. Of the included 24 studies, three studies [33, 43, 47] did not adjust estimated associations for other variables such as age of the children, maternal education, employment of husband, or type of delivery. One study [45] adjusted variables for estimated association with bottle feeding practices but did not adjust for non-exclusive breastfeeding or early initiation of breastfeeding. The recall period of the included studies varied between 24 hours [32] to three years [34]. In one study, the recall period was between 3–15 years [42]. Therefore, the data of the included articles could be affected by recall bias to some extent.

Issues in methodology were identified in some included articles. Two studies [15, 29] defined adequate duration of breastfeeding as up to 36 months and 48 months, respectively; therefore, the effect estimate could have been different if minimum recommended duration by the WHO and UNICEF was considered instead as commonly used in other studies [1]. Two studies [39, 40] were found to have sampling errors. However, the methods of these studies, including the construction of the sampling frame, were not accounted for in detail.

Two studies did not consider maternal employment in their adjusted model, and as such only provided crude results for this variable [45, 47]. A calculation error may have been present in one study [20] whereby the authors gave the prevalence of exclusive breastfeeding at the national level as 35.9% while it is known to be 55% as found in BDHS 2014. This could be a result of including an erroneous number of children under six months of age, caused by the exclusion of mothers who had children less than six months of age, as the denominator when calculating the rate of exclusive breastfeeding.

## Discussion

In this study, we found maternal employment to be a protective factor for recommended complementary feeding practice; however, it appeared as a detrimental factor for early initiation of breastfeeding and exclusive breastfeeding practices. A variation in strength of associations was observed especially when studies were categorized into national and sub-national levels. Other sources of heterogeneity could be due to variations in time of data collection, study setting, sampling techniques, and different variables of interest adjusted for analysis.

Engagement of women in the workforce appears to be a barrier to optimum breastfeeding practices, largely due to inadequate policy support, even though it is an important indicator of development. Similarly, pooled ORs indicate that maternal employment is associated with a higher risk of not engaging in recommended breastfeeding practices. Employed mothers were found to have facilitated access to breast milk substitutes (BMS), which may be detrimental to recommended breastfeeding practices [40]. Lack of policy support, such as the poor implementation of six month mandatory maternity leave, especially in the private sectors, [45, 50–52], lack of available breastfeeding-friendly workplaces [53, 54], lack of provision of adequate breastfeeding breaks [40, 55], and lack of available day care facilities for infants [40, 55] were identified as potential barriers to optimum breastfeeding practices for employed mothers. In addition, cesarean delivery, which has been found to be a barrier to early breastfeeding in Bangladesh [47, 56], was more common among employed mothers, because of their affordability, than unemployed mothers [56, 57]. Meanwhile, unemployed mothers have been found to be more easily reached with information campaigns; therefore, they were more likely to benefit from mainstream nutrition and public health interventions targeting improvements to

breastfeeding knowledge and practices [39, 40]. Considering the aforementioned findings, the maternal employment status in Bangladesh, coupled with suboptimal policy support, seemed to cause working mothers to breastfeed their children for a shorter period [53, 58, 59].

Included studies also explained how maternal employment impacted complementary feeding practices. In general, working mothers were found to have more control over food choices due to improved access through income. In addition, they are more educated and were found to have better knowledge about child health and nutrition than their counterparts, which may lead to better complementary feeding practices [60]. This justification is supported by different studies conducted in Bangladesh, Nepal, and other South Asian countries [61–64]. However, a positive association between maternal employment and better child feeding practices is not always guaranteed. Mothers who were employed but with a low salary and engaged in informal and poor working conditions were at higher risk of failing to secure adequate childcare [65]. Therefore, children whose mothers were working in unsuitable workplace environments, were paid low salaries, or were engaged in the informal sectors experienced poor complementary feeding [16, 35, 36, 46]. These justifications are in accordance with findings from studies further afield, such as in South Africa [66].

In summary, employment can bring a positive impact on the breastfeeding behaviors of mothers. However, it has also been widely found that unsuitable workplace environments with no or insufficient provision of maternity leave act as barriers to the ability of mothers to adhere to recommended breastfeeding practices. Similarly, having a low salary, engaging in informal work, or being subject to poor working conditions appears to be associated with insufficient complementary feeding practices. Therefore, the promotion of breastfeeding and complementary feeding practices through informational and educational interventions cannot alone improve IYCF practices in Bangladesh. The authors recommend appropriately adapting work environments including providing breastfeeding spaces and breaks, raising the minimum wage, and effectively enforcing the international code of breastmilk substitutes could contribute to improving IYCF practices among both employed and unemployed mothers. These initiatives could be beneficial not only for mothers but also for employers: workplace lactation support programs have been shown to improve both breastfeeding outcomes and reduce absenteeism, while also increasing job satisfaction [67].

## Strengths, limitations, and future scopes

To the best of our knowledge, this is the first review to systematically investigate the association of maternal employment with IYCF practices in Bangladesh. Four common databases were selected for literature search. Most of the studies included in the review considered nationally representative data, thereby including large numbers of participants. This ensured greater reliability of the findings of the present study, while also making the results externally generalizable to Bangladesh as a whole. Meanwhile, the high consensus had between the reviewers in the screening process, with Cohen's kappa of inter-rater variability having a mean of 0.86, indicates a low likelihood of selection bias.

The present study was, however, subject to several limitations. To begin with, the associations considered were mostly based on cross-sectional findings, implying that the causality of the association is not guaranteed. As the duration of database search was limited until March 2023, consideration of including any records published beyond that timeframe might result in changes in the pooled associations. Furthermore, several studies using the same dataset were included which may have distorted the apparent weight of certain findings, especially narrative synthesis. When conducting the meta-analysis, reported effect estimates (ORs) were considered rather than synthesizing ORs from the raw data. We examined maternal employment in a

dichotomous manner (yes/no), which did not allow for a detailed understanding of employment conditions and stressors (e.g., flexibility, occupational exposure to chemical, type of contract, atypical schedules etc.). There is a need to better describe the characteristics of maternal employment. Unfortunately, this was rarely done in included studies. Finally, the results of the present study could only be generalized in Bangladesh or similar locations only due to the context-specific nature of the studies included.

According to this meta-analysis, findings from studies conducted at sub-national levels were substantially different from national level data. Therefore, those evidence could be further crosschecked and utilized for context specific intervention design.

## Policy implications and recommendations

There has been a significant improvement in women's engagement in the workforce in the last three decades in Bangladesh; from 9.7% in 1986, to 36.3% in 2017 [24]. This is necessary for economic and national development. Indeed, improvements in income per capita, gross domestic product, and human development index score have observed in Bangladesh over the same period [25]. However, it appeared that maternal employment represented a statistically significant negative association with exclusive breastfeeding. Previous research has identified substantial gaps in the implementation of IYCF policies in Bangladesh despite the presence of 19 policy documents related to IYCF practices [68]. In brief, those policies emphasize the implementation of enabling environments for breastfeeding at the workplace, the promotion and protection of breastfeeding and complementary feeding practices, quality implementation of infant- and child-friendly hospital initiatives, increasing the duration of maternity leave, and mandating the mainstreaming of IYCF [68].

Several suggestions for crucial policies to improve multiple aspects of proper IYCF practices can be made based on the findings of this review. To begin, breastfeeding can be encouraged by ensuring the provision of mandatory maternity leave of at least six months with full compensation in both the public and private sectors. Additionally, constructing adequate working environments allowing for breastfeeding breaks in dedicated spaces may be highly beneficial. Meanwhile, complementary feeding practices must be promoted among unemployed mothers as well as mothers working in unsupportive environments. Several steps could be taken, including boosting levels of employment among women, strengthening existing nutrition interventions, and providing additional IYCF services for mothers of low socioeconomic status. Furthermore, barriers to complementary feeding practices could be removed by adapting workplaces accordingly and ensuring gender-equity in compensation [69]. In general, IYCF programs could be strengthened so that mothers from all socioeconomic conditions could be reached, as appropriate.

## Conclusion

The association between maternal employment and IYCF indicators in Bangladesh does not appear to be homogenous. Maternal employment seemed to be a protective factor for complementary feeding practices but a risk factor for early breastfeeding initiation and exclusive breastfeeding. Support through policy and its implementation should be extended to employed mothers to allow them to practice exclusive breastfeeding during the first six months. Similarly, support should be given to unemployed mothers to allow for improved complementary feeding practices.

## Supporting information

**S1 Appendix. Containing quality score of the included articles, leave-one-out meta-analyses, funnel plots, search strategy, funding sources highlighted in the included articles,**

**PRISMA checklist, list of records excluded during full text review, data extractors and date of data extraction, and dataset for meta-analysis.**
(PDF)

## Acknowledgments

The authors cordially thank Ferdous Akther (FA) for her kind support for accessing Covidence and participating in the title and abstract screening.

## Author Contributions

**Conceptualization:** M. A. Rifat.

**Data curation:** M. A. Rifat, Plabon Sarkar, Israth Jahan Rimu, Syeda Saima Alam, Tasnu Ara.

**Formal analysis:** M. A. Rifat.

**Investigation:** M. A. Rifat, Plabon Sarkar, Israth Jahan Rimu.

**Methodology:** M. A. Rifat, Plabon Sarkar.

**Project administration:** M. A. Rifat, Plabon Sarkar.

**Software:** M. A. Rifat.

**Supervision:** Plabon Sarkar, Sanjib Saha.

**Validation:** M. A. Rifat, Plabon Sarkar, Israth Jahan Rimu, Syeda Saima Alam, Tasnu Ara.

**Visualization:** M. A. Rifat, Plabon Sarkar, Israth Jahan Rimu, Syeda Saima Alam, Tasnu Ara, Tobias Lindström Battle, Manzur Kader, Sanjib Saha.

**Writing – original draft:** M. A. Rifat, Plabon Sarkar.

**Writing – review & editing:** M. A. Rifat, Plabon Sarkar, Israth Jahan Rimu, Syeda Saima Alam, Tasnu Ara, Tobias Lindström Battle, Manzur Kader, Sanjib Saha.

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
