## [Decision Letter · Decision Letter 0]

25 Sep 2023

PONE-D-23-20704How is maternal employment associated with infant and young child feeding in Bangladesh? A systematic literature review and meta-analysisPLOS ONE

Dear Dr. Rifat,

Thank you for submitting your manuscript to PLOS ONE. After careful consideration, we feel that it has merit but does not fully meet PLOS ONE’s publication criteria as it currently stands. Therefore, we invite you to submit a revised version of the manuscript that addresses the points raised during the review process.

ACADEMIC EDITOR: 

You have produced a well considered and presented systematic review, presented in the context of a growing woman workforce in Bangladesh. The three reviewers also recognized the value of this study. Please go through their detailed suggestions carefully and resubmit with minor corrections. In addition to the reviewer comments, I identified a few additional minor points to address in your rebuttal letter:

Line 45, add space between 0.32 and (Line 163, add reference to the Newcastle Ottawa ScaleLine 186, explain rationale for publication periodLine 237, no need to include year of publication when using Vancouver. The same is true later in manuscript, e.g. lines 267, 296, 353. Please check throughout and remove the blue hyperlinksLine 279, mention  that, of the 3 figures, only one was statistically significantLine 361, the UNICEF and WHO recommendations are up to 2 years and beyond. The 'and beyond' is inclusive of these time periods. In other words, longer breastfeeding should not be construed as potentially negative. I understand you were pointing to what the authors deemed 'adequate,' but I suggest looking at how this sentence is written.Lines 383-4, specify what current maternity policies are in Bangladesh (this can be presented earlier if desired)Line 288, provide further background on existing nutrition efforts (can be presented earlier in background)==============================

We look forward to receiving your revised manuscript.

Kind regards,

Sara Jewett Nieuwoudt, Ph.D, MPH

Academic Editor

PLOS ONE

.

Reviewers' comments:

Reviewer's Responses to Questions

**Comments to the Author**

1. Is the manuscript technically sound, and do the data support the conclusions?

Reviewer #1: Yes

Reviewer #2: Yes

Reviewer #3: Yes

2. Has the statistical analysis been performed appropriately and rigorously? 

Reviewer #1: Yes

Reviewer #2: I Don't Know

Reviewer #3: No

3. Have the authors made all data underlying the findings in their manuscript fully available?

Reviewer #1: Yes

Reviewer #2: Yes

Reviewer #3: No

4. Is the manuscript presented in an intelligible fashion and written in standard English?

Reviewer #1: Yes

Reviewer #2: Yes

Reviewer #3: Yes

5. Review Comments to the Author

Reviewer #1: Thank you for inviting me to read this interesting literature review.

Feedback:

1) Introduction, Line 60: Could the authors provide some context on how the statistics quoted compared to other similar countries? This will help readers understand the position of Bangladesh within the global context.

2) Introduction, Line 72: The reference used (#9) is about Ethiopia, not Bangladesh. Can the authors replace this with a reference relevant to the study population? If not, perhaps rework the sentence, for example, “Researchers have shown in LMIC employed mothers …”

3) Methods, Lines 126-129: These two sentences seem to contradict each other on the data extraction process. Can the authors please review and edit to ensure clarity?

4) Methods, Lines 139-140: The authors introduce MDD and MAD with little explanation of this terminology. Can the authors please provide more information about this terminology?

5) Methods: Can the authors clarify whether the included papers defined the duration of exclusive breastfeeding in line with WHO recommendations?

6) Results, Line 208: I cannot see a Table 3. Does Table 4 need to be renumbered as Table 3?

7) Results, Lines 218-271: Much of this 2.5 pages of written content repeats the results presented in the forest plots. Can the authors condense the results by briefly commenting on the frequently contradictory nature of the included papers’ results, and then focussing on the pooled estimates in the written text, rather than repeating results for individual studies?

8) Discussion, Lines 318-9: The authors state ‘working mothers were found to have more control over food choices due to improved access through income” however it is not clear to me that the data presented in the Results section supports this statement. Can the authors please provide a clear link between the results and this statement?

9) Discussion, Lines 322-25: It is unclear whether the authors are referring to their own results or results from other papers. The authors need to reframe their writing to make this clear to the author and use correct tense (were vs. are). For example, “In contrast, it has been shown mothers who are employed …. are at higher risk …”

10) Discussion, Line 368: Can the authors clarify this potential calculation error in more detail and how they came to determine this potential error explanation may have occurred?

11) Discussion, Line 370: The Policy Implications Section reads as if it should come at the end of the paper, leading into the Conclusions. I would recommend the authors consider moving the Limitations Section to after the Study Quality but before the Policy Implications Section. This will improve the flow of the paper.

12) Discussion, Line 408: The authors introduce abbreviations (EIBF, EBF, CF) which have not been previously defined or used elsewhere in the paper. Can the authors either write the abbreviations out in full, or define these earlier and use throughout?

General grammar feedback:

- Numbers under 10 should be written as words, not numerals.

- Blue font appears in some sections of the paper.

Reviewer #2: The data processing seems in order, the conclusion can be re-written as the current conclusion is too broad and weak, given the strength of the evidence presented. Strengthen to conclusion to speak directly to the findings. Employed women were found to be less likely to initiate breastfeeding, this calls for improved maternity protection and support for employed women. Employed women were less likely to exclusively breastfeed, this calls for stronger workplace support and breastfeeding protection. CF support needs to be implemented for all children, irrespective of the mother's employment status.

Reviewer #3: Title of the manuscript: How is maternal employment associated with infant and young child feeding in Bangladesh? A systematic literature review and meta-analysis

Number of the manuscript: PONE-D-23-20704.

Reviewer: Christian B. Ngandu

Date: September 21, 2023.

Reviews comments

General comments

• The paper is well designed and very well written. There is a consistency between the study aim with sections of the manuscript. The sections of the manuscript are consistent to the study aim and written with clarity and concision. The systematic review process has been followed and realized. However, the manuscript should be edited and more details on the results sections are needed.

• Methods section should be strengthened by in-text citations. The meaning of effects size estimates and their significance should be verified in the whole text.

Specific comments

Abstract

• Lines 29-30. The authors should pay attention to the accuracy at citing the variables in the study aim. Please, rephrase that section of the sentence: ‘…to investigate the association between these variables.

• Line 34. This sentence ‘Findings were presented in a summary table’ doesn’t add any value to the abstract, I will suggest the authors delete it.

• Many studies have been conducted on the topic. The novelty of the manuscript might be to mention the study period and the study setting to show the specificity of the study.

Introduction

• Lines 58-60. Replace the in-text citation to the end of the sentence to support the whole sentence.

• Line 85. The authors should rephrase this sentence: ‘…between increases in maternal employment and IYCF practices has yet to be reviewed.’ I suggest: ‘…between in maternal employment status and IYCF practices has yet to be reviewed.’

• Line 87. The metanalysis provides the consensus on the associated pooled data and not a spurious idea on the relationship. The authors can use the term “consensus” instead of “an idea”.

Methodology

• Selection process section (Lines 114-123). The manuscript seems to have two eligibility criteria: the first one on lines 103-111 and the second on line 118-123. We should recommend the authors to use one eligibility criteria based on the PECO acronym for the whole text. Also, this eligibility criteria should include the period where data has been collected.

• Please, edit the format of the sentence: ‘… title and abstract were…’ instead of ‘… title and abstract was…’

• Data extraction section. The are many resources (citations) that can support the data extraction and very interesting o be cited. Among the variables that have been extracted ion the data extraction table, the title of each primary article doesn’t need to display.

• Data synthesis and metanalysis section.

o the section should need more in-text citations to support the review process particularly for data synthesis, meta-analysis, funnel plot.

o Criteria for including primary articles in the meta-analysis needs to be clearly defined in this section. For instance, the definition of variables were they the same for all primary articles that have been included in the metanalysis?

• Quality assessment and risk assessment.

o This section should chronologically display after the data synthesis before the metanalysis. The quality of data is one of the criteria for review.

Results

• Figure 1:

o the computation of numbers in the whole flow diagram needs more correctitude and consistency. All databases have searched for 1865 primary articles instead of 1971 that have been mentioned in the flow diagram. The authors should spell out the search strategy in the manuscript for transparency and reproducibility of the methods. Also, the citation for the flow diagram can be inserted in the figure comments.

o the cell ‘Studies for retrieval’ should display 67 articles instead of 65 as reflected in the flow diagram.

o the reasons of exclusion of primary articles need to be mentioned for more accuracy in the review process. ‘wrong exposure’ or ‘wrong outcomes’ seem to be inappropriate terms to express the exclusion criteria. Terms as ‘not relevant exposure or not relevant outcomes’ refer to the eligibility criteria and are a bit more appropriate.

• Figure 2 to 4:

o Only one study (Hans, 2020) has demonstrated a protective relationship between maternal employment status and the IYCF.

o the summary of the review should better explore these sections. The pooled data should better be summarized in the first paragraph of the discussion instead of the summary from primary articles. For illustration, data can show the positive association between exclusive breastfeeding, complementary feeding and maternal employment status while no significant relationship between maternal employment and initiative breastfeeding.

• Table 1:

o the heading in the table 1, column 2 need to be edited. Please, can the authors mention ‘Author, year’ instead of ‘study’?

o for better understanding of the table 1, I would recommend the authors to add another column beside of the entitled column ‘geographical area variable’. It will better sound to segregate data based on ‘National vs subnational’ for the first column, and the second column where ‘rurality variable’ should include ‘rural vs urban vs urbano-rural’.

o Demographic data for participants (mother/caregiver, children) might add value to the review.

o On line 264, how confident are the authors to the results? Please, check for the confidence interval? Thew authors should check for the significance of the relationship between maternal employment status and IYCF. Check for the effects size estimates on lines 224, 237, 260, and 278.

• Limitations section. On lines 401-404, four databases that were included in the review stand for a strength of the review instead of the weakness.

References

• Repetition of the websites for the first citation number 1 should be avoided.

• The format of references needs some edited. The use of capital letter for the titles of citations needs to be edited: citations number 17, 19, 23, 28 and 30.

6. PLOS authors have the option to publish the peer review history of their article (what does this mean?). If published, this will include your full peer review and any attached files.

Reviewer #1: No

Reviewer #2: **Yes: **Chantell Witten

Reviewer #3: No

---

## [Author Response · Author response to Decision Letter 0]

18 Nov 2023

Thank you very much for providing us with an opportunity to revise the manuscript. We have addressed each of the comments carefully and edited the manuscript accordingly. The response to editor’s and reviewers’ comments have been provided on a point-by-point basis in the attached “Response to Reviewers” file. Please inform us if any further clarification or revision is required. We are eagerly looking forward to your response.

---

## [Decision Letter · Decision Letter 1]

16 Apr 2024

PONE-D-23-20704R1How is maternal employment associated with infant and young child feeding in Bangladesh? A systematic literature review and meta-analysisPLOS ONE

Dear Dr. Rifat,

Thank you for submitting your manuscript to PLOS ONE. After careful consideration, we feel that it has merit but does not fully meet PLOS ONE’s publication criteria as it currently stands. Therefore, we invite you to submit a revised version of the manuscript that addresses the points raised during the review process.

**ACADEMIC EDITOR: **I would firstly like to thank you for your patience. Securing reviewers proved to be a challenging task after several original reviewers indicated unavailability. This systematic review provides valuable insights for the Bangladesh context about how employment may impact on breastfeeding and complementary feeding. For the most part, an appropriate level of detail is provided around the methodology to enable repetition. That said, there are a few minor corrections that are required. Regarding Reviewer 1's comments specific to updating the search and the timing of data collection, as long as the endpoint of the search is clearly denoted and timing is noted in limitations, there is no expectation that you need to re-do the search and update the analysis as this late stage in the manuscript review process. Similarly, regarding the suggestion on Table 1, you can note the study design in the first column instead of creating an entirely new column if you face space constraints.    Once the remaining points are addressed, we should be able to proceed.

We look forward to receiving your revised manuscript.

Kind regards,

Sara Jewett Nieuwoudt, Ph.D, MPH

Academic Editor

PLOS ONE

Journal Requirements:

Reviewers' comments:

Reviewer's Responses to Questions

**Comments to the Author**

1. If the authors have adequately addressed your comments raised in a previous round of review and you feel that this manuscript is now acceptable for publication, you may indicate that here to bypass the “Comments to the Author” section, enter your conflict of interest statement in the “Confidential to Editor” section, and submit your "Accept" recommendation.

Reviewer #1: (No Response)

Reviewer #2: (No Response)

2. Is the manuscript technically sound, and do the data support the conclusions?

Reviewer #1: Partly

Reviewer #2: Yes

3. Has the statistical analysis been performed appropriately and rigorously? 

Reviewer #1: I Don't Know

Reviewer#2: Yes

4. Have the authors made all data underlying the findings in their manuscript fully available?

Reviewer #1: Yes

Reviewer #2: Yes

5. Is the manuscript presented in an intelligible fashion and written in standard English?

Reviewer #1: Yes

Reviewer #2: Yes

6. Review Comments to the Author

Reviewer #1: Methods, Search Strategy, Line 100: The database searches are now out of date (over 12 months ago). These need to be updated prior to publication. In addition, academic rigor requires all searches to be performed on the same date.

Methods, Inclusions and Exclusion Criteria, Lines 102-111: Most of this section would be better presented in a table. The last sentence is incomplete.

Results, Meta-analysis, Line 247: The wording of this sentence is unclear. Do the authors mean four studies provided data for inclusion in the effect size estimates for early initiation of breastfeeding? Similarly, my query also relates to the wording structure of the rest of the sentence.

Results, Meta-analysis, Line 249: Can the authors confirm whether original extracted data from the papers were used to calculate the pooled estimates? This is best practice, rather than using reported OR/RR/HR to calculate pooled estimates.

Results, Table 1: Since the study includes different types of observational studies, a column in the table should be included to indicate this information about each included paper.

Formatting:

Please check the font throughout the discussion as there are some inconsistencies, mostly occurring at the insertion points for references.

Please ensure the manuscript adheres to standard English conventions, including the writing of numbers under 10 as words, and numbers over 10 as numerals, e.g. Line 247, ‘4 effect sizes’ should be ‘four effect sizes’.

Reviewer #2: This is a strong manuscript that merits publication. The following suggestions are shared to strengthen a few points and to address proofreading errors.

1. The pooled analysis study results need to state more overtly that only EBF was statistically significant, whereas there was not a statistically significant difference by employment status for the other IYCF outcomes of interest. While p-values are now included, there is still a need to make this point clearly. The sentences that report results should signal this in the abstract and meta-analysis section. The first sentence of the discussion (lines 268-9) and also lines 372-3 should be reviewed and revised. The discussion of the divergent findings between studies is done well.

2. For data synthesis when ORs were not used, specify what was used to calculate them into ORs. If this was not done using raw data, this needs to be noted as a limitation

3. Check pagination. After page 8, the page numbers restarted at 1

4. Address wording in the following:

-line 220, change 'rest one study' to 'remaining study'

-line 241-2, change 'private job' to 'private sector job'

-line 256, change 'week' to 'weak'

5. Figure 3 is missing the red dashed reference line at 1 (this was included for Figures 2 and 4)

6. On line 278, what was the basis for expecting that breastfeeding outcomes would be better for employed mothers? Most global literature has found consistently that employment is a barrier to EBF (see 2016 Lancet series, for instance). Consider rewriting this sentence in a more nuanced way, to reflect how literature on whether employment facilitates or hinders optimal IYCF depending on the outcome of interest. This is done well in the introduction section of the manuscript.

7. My final question is about the location of the quality assessment section. I thought this would be better placed in the results section vs discussion. The limitations section can then reiterate any concerns in the discussion section. However, the authors can be guided by conventions and preferences regarding this point.

7. PLOS authors have the option to publish the peer review history of their article (what does this mean?). If published, this will include your full peer review and any attached files.

Reviewer #1: No

Reviewer #2: No

---

## [Author Response · Author response to Decision Letter 1]

17 May 2024

Rear reviewers and editor, 

Thank you very much for you insightful comments and suggestions. These were highly useful to improve the scientific rigor of the manuscript. We have responded each of the comments you provided. The responses are presented in "Response to reviewers" file attached with the revised manuscript. Please let us know if any changes in the manuscript and clarification is required. 

Sincerely,

Corresponding author

---

## [Decision Letter · Decision Letter 2]

26 Jul 2024

PONE-D-23-20704R2How is maternal employment associated with infant and young child feeding in Bangladesh? A systematic literature review and meta-analysisPLOS ONE

Dear Dr. Rifat,

Thank you for submitting your manuscript to PLOS ONE. After careful consideration, we feel that it has merit but does not fully meet PLOS ONE’s publication criteria as it currently stands. Therefore, we invite you to submit a revised version of the manuscript that addresses the points raised during the review process.

**The comments of reviewer are described below. Although the reviewer classified all changes as "minor revision", I suggest accepting all of them to improve the clarity and quality of the manuscript. ** Please submit your revised manuscript by Sep 09 2024 11:59PM. If you will need more time than this to complete your revisions, please reply to this message or contact the journal office at plosone@plos.org. Please include the following items when submitting your revised manuscript:A rebuttal letter that responds to each point raised by the academic editor and reviewer(s). You should upload this letter as a separate file labeled 'Response to Reviewers'.A marked-up copy of your manuscript that highlights changes made to the original version. You should upload this as a separate file labeled 'Revised Manuscript with Track Changes'.An unmarked version of your revised paper without tracked changes. You should upload this as a separate file labeled 'Manuscript'.If applicable, we recommend that you deposit your laboratory protocols in protocols.io to enhance the reproducibility of your results. Protocols.io assigns your protocol its own identifier (DOI) so that it can be cited independently in the future. For instructions see: https://journals.plos.org/plosone/s/submission-guidelines#loc-laboratory-protocols. Additionally, PLOS ONE offers an option for publishing peer-reviewed Lab Protocol articles, which describe protocols hosted on protocols.io. Read more information on sharing protocols at https://plos.org/protocols?utm_medium=editorial-email&utm_source=authorletters&utm_campaign=protocols.

We look forward to receiving your revised manuscript.

Kind regards,

Ana Elisa Madalena Rinaldi, Ph.D.

Academic Editor

PLOS ONE

Journal Requirements:

Reviewers' comments:

Reviewer's Responses to Questions

**Comments to the Author**

1. If the authors have adequately addressed your comments raised in a previous round of review and you feel that this manuscript is now acceptable for publication, you may indicate that here to bypass the “Comments to the Author” section, enter your conflict of interest statement in the “Confidential to Editor” section, and submit your "Accept" recommendation.

Reviewer #4: (No Response)

Reviewer #5: All comments have been addressed

2. Is the manuscript technically sound, and do the data support the conclusions?

Reviewer #4: Yes

Reviewer #5: Yes

3. Has the statistical analysis been performed appropriately and rigorously? 

Reviewer #4: Yes

Reviewer #5: Yes

4. Have the authors made all data underlying the findings in their manuscript fully available?

Reviewer #4: Yes

Reviewer #5: Yes

5. Is the manuscript presented in an intelligible fashion and written in standard English?

Reviewer #4: Yes

Reviewer #5: Yes

6. Review Comments to the Author

Reviewer #4: I have with great interest read the manuscript "How is maternal employment associated with infant and young child feeding in Bangladesh? A systematic literature review and meta-analysis". The manuscript is easy to read and was conducted through a rigorous method with the use of validated tools. This work addresses an important issue regarding the development of healthy feeding behavior in infant. Above all, I particularly appreciated the approach based on a socio-structural determinant of health (maternal employment) which is the most promising way to improve the health of populations and fight against social inequalities in health. This last aspect deserves to be highlighted in the manuscript. The authors propose concrete courses of action for public authorities. For these reasons, I approve the publication of the manuscript, with the following minor revisions considered.

Introduction

Start of introduction: Given that PLOS ONE is a general journal covering all domains of science and not specific to infant feeding behaviors, it would be useful to briefly remind readers of the WHO recommendations regarding breastfeeding and complementary feeding.

Line 58. “In Bangladesh, 65% of infants aged less than 6 months are exclusively breastfed…”. This 65% value remains relatively high compared to other countries and is even higher than the global exclusive breastfeeding rate, which reached 48% in 2023 according to the Global Breastfeeding Scorecard from UNICEF and WHO. To better understand breastfeeding practices in Bangladesh, I suggest comparing this rate with those of other low- and lower middle-income countries. Another option would be to discuss the evolution of the exclusive breastfeeding rate in Bangladesh: Exclusive breastfeeding declines with age, from 85% among children age 0-1 months to 40% among those age 4-5 months (ref n°2) + exclusive breastfeeding rate at 6 months if you have this information. This wording would be more representative.

Lines 81-83. “Sub-optimum policy support related to maternity leave or lack of infant-friendly workplaces, including breastfeeding breaks, can lead to poor breastfeeding practices among employed mothers (18).” Considering the major influence of maternal leave policy and the close relationship with maternal employment status, it seems important to give more details of the specific situation in Bangladesh: how long is maternity leave? Are they well-paid? Is there parental leave (gender equity)?

Line 91-93. “This could facilitate the implementation of IYCF policies and programs for both employed and unemployed mothers in Bangladesh to improve IYCF practices.” As mentioned above, addressing IYCF practices regarding maternal employment status (which is a socio-structural determinant of health) is a particularly relevant approach to reducing social inequalities in health. This aspect deserves to be highlighted in introduction and/or conclusion.

Methods

Inclusion and exclusion criteria:

Line 104. “population (P) = mothers having an infant or a young child in Bangladesh”. What do you mean by ‘young child’, have you set an age limit?

“exposure (E) = mothers being employed in any public or private sector jobs”. Did you exclude independent jobs? (self-employed)

“outcome (O) = status of IYCF practices” Not clear enough. Please provide some example in brackets (breastfeeding status, food offered, diversity, quality … )

Line 108. “…between the maternal occupation and IYCF practices”. Please replace ‘occupation’ by ‘employment status’ to be homogenous

Line 109. “There was no limit on the publication period”. The introduction mentions changes in socioeconomic status of the population and women’s participation in the workforce over the last three decades. This could have been an argument for selecting articles since 1990. Why not set a date limit? Does the 1981 study (Jain) remain interpretable given the different context?

Results

Line 220. “Of the three studies investigating the association of maternal employment and breastfeeding duration ». It seems that article n°30 (Ayesha) have measured breastfeeding duration (maybe exclusive and/or mixed) and not only exclusive breastfeeding duration. In this case, please specify “..maternal employment and any breastfeeding duration” to be in agreement.

Line 233. “… employed mothers were more likely to provide complementary feeding than their unemployed counterparts”. This is not clear enough. Do you mean to provide adequate complementary feeding? Please add details.

Line 239. “… with government jobs demonstrated better performance than mothers having private sector jobs”. Similarly, this sentence is not clear enough, what do you mean by ‘better performance’, please provide some details.

Discussion

Line 301. “In this review we found maternal employment to be both a protective and detrimental factor of recommended IYCF practices in Bangladesh depending on the IYCF components”. The first sentence of the discussion is expected to precisely summarize the results. It would be better to specify the 'IYCF components': breastfeeding initiation and duration, MDD, and MAD. Also, please remain consistent in terms, IYCF components/indicators.

Line 319. “In addition, cesarean delivery, which has been found to be a barrier to early breastfeeding in Bangladesh (43,54), was more common among employed mothers than unemployed mothers”. Do you have any explanation for this result? If so, please provide some details to better understand the mechanisms.

Line 324 “Considering the aforementioned findings, the conditions of employment in Bangladesh, coupled with suboptimal policy support…” Please replace “the conditions of employment” by “the maternal employment status”. Indeed, your review does not focus on employment conditions (e.g., job insecurity, flexibility, hours, physical strain etc.) because most of the selected studies explore employment in a dichotomous manner (working yes/no).

Line 328. “In general, working mothers were found to have more control over food choices due to improved access through income. In addition, they were found to have better knowledge about child health and nutrition, which may lead to better complementary feeding practices”. What about the education level? This is known to be a strong determinant of health, and I suppose that Bangladeshi employed mothers are more educated than their unemployed counterparts. As you mentioned, this may imply a higher degree of health awareness, better health literacy, more resources to seek help, and better compliance with the existing recommendations but also access to high-skill jobs that promote work-life balance.

Strengths, limitations, and future scopes

I suggest adding a methodological limitation: the fact that the studies explore maternal employment in a dichotomous manner, which does not allow for a detailed understanding of employment conditions and stressors (e.g., flexibility, occupational exposure to chemical, type of contract, atypical schedules etc.). There is a need to better describe the characteristics of maternal employment. Unfortunately, this is rarely done in studies.

Reviewer #5: I believe the authors have address the reviewer comments and that further delays in publication are not necessary. That said, in addition to a final round of proofreading, I have a few suggestions the authors may want to consider.

1. In the abstract, the final sentence could be more specific about breastfeeding-friendly workplace policies for employed mothers and could specify what policies may support unemployed mothers.

2. On p4, line 81, I wonder if the correct term is infant-friendly workplaces or breastfeeding-friendly? I'm more familiar with the latter, but this is not a sticking point.

3.Either under the terminologies section or under each result area, consider adding the technical definitions of the different IYCF practices, such as early initiation, EBF, etc.

4. Line 237, mention of 'association' should be 'associations'

5. Line 272, you can also cross-reference Table 1 now for the NOS scores, in addition to S1

Congratulations to the authors on an interesting paper that belongs in the public domain.

7. PLOS authors have the option to publish the peer review history of their article (what does this mean?). If published, this will include your full peer review and any attached files.

Reviewer #4: **Yes: **Pauline Brugaillères

Reviewer #5: No

---

## [Decision Letter · Decision Letter 3]

11 Dec 2024

How is maternal employment associated with infant and young child feeding in Bangladesh? A systematic literature review and meta-analysis

PONE-D-23-20704R3

Dear Dr. Rifat,

We’re pleased to inform you that your manuscript has been judged scientifically suitable for publication and will be formally accepted for publication once it meets all outstanding technical requirements.

Kind regards,

Jennifer Yourkavitch

Academic Editor

PLOS ONE

Additional Editor Comments (optional):

Reviewers' comments:

Reviewer's Responses to Questions

**Comments to the Author**

1. If the authors have adequately addressed your comments raised in a previous round of review and you feel that this manuscript is now acceptable for publication, you may indicate that here to bypass the “Comments to the Author” section, enter your conflict of interest statement in the “Confidential to Editor” section, and submit your "Accept" recommendation.

Reviewer #5: All comments have been addressed

2. Is the manuscript technically sound, and do the data support the conclusions?

Reviewer #5: Yes

3. Has the statistical analysis been performed appropriately and rigorously? 

Reviewer #5: N/A

4. Have the authors made all data underlying the findings in their manuscript fully available?

Reviewer #5: Yes

5. Is the manuscript presented in an intelligible fashion and written in standard English?

Reviewer #5: Yes

6. Review Comments to the Author

Reviewer #5: I look forward to seeing this in print. Congratulations for this contribution to our understanding of how employment may influence infant feeding in the Bangladesh context.

7. PLOS authors have the option to publish the peer review history of their article (what does this mean?). If published, this will include your full peer review and any attached files.

Reviewer #5: **Yes: **Sara Jewett

---

## [Editor Report · Acceptance letter]

30 Dec 2024

PONE-D-23-20704R3 

PLOS ONE

Dear Dr. Rifat, 

I'm pleased to inform you that your manuscript has been deemed suitable for publication in PLOS ONE. Congratulations! Your manuscript is now being handed over to our production team.

Kind regards, 

on behalf of

Dr. Jennifer Yourkavitch 

Academic Editor

PLOS ONE